# Circulating microRNA/isomiRs as novel biomarkers of esophageal squamous cell carcinoma

Yuta Ibuki[1,2], Yukie Nishiyama[1], Yasuhiro Tsutani[2], Manabu Emi[2], Yoichi Hamai[2], Morihito Okada[2], Hidetoshi Tahara[1,3,4] *

**1** Department of Cellular and Molecular Biology, Graduate School of Biomedical and Health Science, Hiroshima University, Hiroshima, Japan, **2** Department of Surgical Oncology, Research Institute for Radiation Biology and Medicine, Hiroshima University, Hiroshima, Japan, **3** Collaborative laboratory of Liquid Biopsy, Graduate School of Biomedical and Health Sciences, Hiroshima University, Hiroshima, Japan, **4** The Research Center for Drug Development and Biomarker Discovery, Hiroshima University, Hiroshima, Japan

\* toshi@hiroshima-u.ac.jp

## Abstract

**Data Availability Statement:** All relevant data are within the paper and its Supporting Information files.

### Background

MicroRNA (miR)s are promising diagnostic biomarkers of cancer. Recent next generation sequencer (NGS) studies have found that isoforms of micro RNA (isomiR) circulate in the bloodstream similarly to mature micro RNA (miR). We hypothesized that combination of circulating miR and isomiRs detected by NGS are potentially powerful cancer biomarker. The present study aimed to investigate their application in esophageal cancer.

### Methods

Serum samples from patients with esophageal squamous cell carcinoma (ESCC) and age and sex matched healthy control (HC) individuals were investigated for the expression of miR/isomiRs using NGS. Candidate miR/isomiRs which met the criteria in the 1st group (ESCC = 18 and HC = 12) were validated in the 2nd group (ESCC = 30 and HC = 30). A diagnostic panel was generated using miR/isomiRs that were consistently confirmed in the 1st and 2nd groups. Accuracy of the panel was tested then in the 3rd group (ESCC = 18 and HC = 18). Their use was also investigated in 22 paired samples obtained pre- and post-treatment, and in patients with esophageal adenocarcinoma (EAD) and high-grade dysplasia (HGD).

### Results

Twenty-four miR/isomiRs met the criteria for diagnostic biomarker in the 1st and 2nd group. A multiple regression model selected one mature miR (miR-30a-5p) and two isomiRs (isoform of miR-574-3p and miR-205-5p). The index calculated from the diagnostic panel was significantly higher in ESCC patients than in the HCs (13.3±8.9 vs. 3.1±1.3, p<0.001). The area under the receiver operating characteristics (ROC) curves of the panel index was 0.95. Sensitivity and specificity were 93.8%, and 81% in the 1st and 2nd groups, and 88.9% and

**Funding:** The funder (MiRTeL Co. Ltd) provided support in the form of salaries for an author (H.T), but did not have any additional role in the study design, data collection and analysis, decision to publish, or preparation of the manuscript. The specific roles of these authors are articulated in the 'author contributions' section.

**Competing interests:** Prof. Hidetoshi Tahara is representative director of a university-originated venture, MiRTeL Co. Ltd., which provides commercial microRNA panel services. The remaining authors declare no potential conflict of interest. MiRTeL Co. Ltd provided support in the form of salaries for an author (H.T), but did not have any additional role in the study design, data collection and analysis, decision to publish, or preparation of the manuscript. This does not alter our adherence to PLOS ONE policies on sharing data and materials.

72.3% in the 3rd group, respectively. The panel index was significantly lower in patients with EAD (6.2±4.5) and HGD (4.2±1.7) than in those with ESCC and was significantly decreased at post-treatment compared with pre-treatment (6.2±5.6 vs 11.6±11.5, p = 0.03).

## Conclusion

Our diagnostic panel had high accuracy in the diagnosis of ESCC. MiR/isomiRs detected by NGS could serve as novel biomarkers of ESCC.

## Introduction

Esophageal cancer is one of the most common cancers worldwide and has high mortality [1, 2]. The prognosis of patients with esophageal cancer remains poor despite recent improvements in therapy and perioperative management, and 5-year survival rate remains about 20%, even in developed countries [3]. One reason for this poor prognosis is that most patients with esophageal cancer are diagnosed at an advanced stage [4]. In contrast, early stage esophageal cancer, in particular mucosal cancer is expected cure by endoscopic resection [5, 6]. This substantial discrepancy suggests that a specific diagnostic biomarker could be used for early detection would improvement the prognosis of patients with esophageal cancer. While several biochemical markers have been investigated, including squamous cell carcinoma antigen [7], carcinoembryonic antigen [8] and, CYFRA 21–1 [9], their sensitivity has not proved consistently satisfactory across the various stages of esophageal cancer.

MicroRNA(miR)s are classified as small noncoding RNAs (19–25 nucleotides) which regulate the expression of plural numbers of messenger RNAs [10–12]. Cancer cells possess miRs which have particular function in promoting cancer development or minimizing cancer suppression. miRs also exist in the blood stream as inclusions in exosomes. These circulating miRs play a role in intercellular communication in the cancer environment and bring about favorable conditions for cancer invasion and metastasis. Because their expression profiles vary between cancer patients and healthy individuals, circulating miRs can act as powerful biomarkers in the diagnosis of cancer. Indeed, many researchers have reported their usefulness as novel biomarkers for several malignant tumors, including esophageal cancer [13–17].

Recent research from deep sequencing represented by the next generation sequencer(NGS) has revealed that miRs are heterogeneous. Isoforms of miR differ slightly from mature miR by base length and sequence and are referred to as isomiR. Although the function of isomiR is not completely understood, they are known to play an important role in cancer development [18, 19]. IsomiRs also exist in the blood with high stability, similarly to mature miRs. We hypothesized that combination of circulating miR and isomiRs detected by NGS might act as novel biomarkers for malignant tumors. To date, however, few studies examined the usefulness of miR/isomiRs from blood samples as cancer biomarkers. Here, we aimed to investigate their application in esophageal squamous cell carcinoma (ESCC) using NGS.

## Material and methods

### Samples

We prospectively collected serum samples of patients treated for esophageal cancer at Hiroshima University Hospital from January 2010 to July 2018. Before April 2016, samples for patients undergoing surgery were collected only at surgery. Thereafter, samples were collected

before treatment from all patients with esophageal cancer, such as at endoscopic resection, chemoradiotherapy, neoadjuvant therapy followed by surgery, and palliative chemotherapy. We used 18 consecutive samples from January 2010 to December 2012, 30 from January 2013 to February 2017, and 18 from March 2017 to July 2018 as the first (1[st]), second (2[nd]), and third (3[rd]) groups, respectively. Healthy control (HC) samples were collected at the same time by our laboratory from individuals who were confirmed not to have a medical history of cancer. Among them, 12, 30, and 18 samples were enrolled in the 1[st], 2[nd], and 3[rd] groups, with consideration to matching sex and age with ESCC patients. Table 1 summarizes the characteristics of patients and HCs. All patients were histologically diagnosed with squamous cell carcinoma and staged according to the 8[th] Edition of the TNM Classification of Malignant Tumors [20]. Treatment strategy was determined at our institutions according to clinical stage and patient condition as described previously [21, 22]. Briefly, mucosal cancer was treated with endoscopic resection, submucosal cancer without lymph node metastasis with initial surgery; and respectable advanced cancer with neoadjuvant therapy followed by surgery if overall patient condition was good. Patients who did not wish to undergo surgery or judged unsuitable for resection were treated with definitive chemoradiotherapy, while those with distant metastasis were given palliative chemotherapy.

Among the 66 samples from patients with ESCC before treatment, 22 were collected at 1 month after treatment. Serum samples were also collected from 4 patients who experienced postoperative recurrence at the time of recurrence. Furthermore, samples were collected from patients with esophageal adenocarcinoma (EAD; n = 4) and high-grade dysplasia (HGD; n = 4) who were enrolled to assess specificity for ESCC. Fig 1 shows overview of this study. The study was approved by the Institutional Review Board of Hiroshima University.

### RNA extraction from serum samples

After obtaining informed consent, 2ml of peripheral blood was obtained from each patient before any treatment procedure.Serum was separated by centrifugation at 3000 rpm for 10 min at 4˚C. The supernatant was collected into a new tube and the serum sample was stored at -80˚C. Total RNA was isolated from 200 μl serum using a miRNeasy mini kit (Qiagen) according to the manufacturer's protocol.

### cDNA library for micro RNA sequencing

An Ion Total RNA-Seq Kit v2 was used to prepare a reconstructive cDNA library for preparation of small RNA sequencing. The size and concentration of base pairs of the cDNA library were measured with an Agilent 2100 Bioanalyzer (Agilent Technologies). Preparation for deep sequencing such as emulsion PCR, bead enrichment, and chip loading were automatically performed on an Ion Chef– instrument (Thermo Fisher Scientific). In the final step of sample preparation for sequencing, the chip was loaded with the Ion Sphere Particle (ISP) sequencing reaction mixture. Synthesized templates were sequenced on an Ion S5–XL sequencer (Thermo Fisher Scientific) using an Ion 540– chip.

### Data analysis

After the sequencing reaction, the data were checked for quality. We defined acceptable data as 70% or more above ISP loading density, and 60 or more templates per ISP; 30% or more usable reads, and 5% or less test fragments per total reads; and 100000 or more usable reads per sample. Acceptable data was analyzed using a CLC genomics work bench 7(CLC bio). Small RNAs were merged by count read number and annotated based on miRbase version 21 (http://www.mirbase.org/). IsomiRs were identified by differences such as additions or

**Table 1.** **A.** Characteristics in 1$^{st}$ group. **B.** Characteristics in 2$^{nd}$ group. **C.** Characteristics in 3$^{rd}$ group.

| A | | |
|---|---|---|
| | ESCC patients (n = 18) | Control (n = 12) |
| Age (y) | 68 (54–85) | 62 (50–83) |
| Sex | | |
| Male | 13 (72.2%) | 9 (75%) |
| Female | 5 (28.8%) | 3(25%) |
| Smoking | | |
| None | 4 (22.2%) | |
| Ex | 6 (33.3%) | |
| Current | 8 (44.4%) | |
| Tumor location | | |
| Ut | 7 (38.9%) | |
| Mt | 6 (33.3%) | |
| Lt | 3 (16.7%) | |
| Ae | 2 (11.1%) | |
| Differentiation | | |
| Well | 7 (38.9%) | |
| Moderate | 9 (50%) | |
| Poor | 2 (11.1%) | |
| Unknown | 0 | |
| Clinical Stage | | |
| I | 15 (83.3%) | |
| II | 2 (11.1%) | |
| III | 1 (5.6%) | |
| IVA | 0 | |
| IVB | 0 | |
| Treatment | | |
| ESD | 0 | |
| Surgery | 18 (100%) | |
| Neoadjuvant therapy | 0 | |
| CRT | 0 | |
| CT | 0 | |
| Pathological Stage | | |
| IA | 5 (27.8%) | |
| IB | 10 (55.6%) | |
| IIA | 1 (5.6%) | |
| IIB | 1 (5.6%) | |
| IIIA | 0 | |
| IIIB | 1 (5.6%) | |
| IVA | 0 | |
| IVB | 0 | |
| B | | |
| | ESCC patients (n = 30) | Control (n = 30) |
| Age (y) | 69 (54–80) | 66(55–77) |
| Sex | | |
| Male | 25 (83.3%) | 23 (76.7%) |
| Female | 5 (16.7%) | 7 (23.3%) |
| Smoking | | |

(*Continued*)

**Table 1.** (Continued)

| | | |
|---|---|---|
| None | 4 (13.3%) | |
| Ex | 9 (30%) | |
| Current | 17 (56.7%) | |
| Tumor location | | |
| Ut | 6 (20%) | |
| Mt | 10 (33.3%) | |
| Lt | 14 (46.7%) | |
| Ae | 0 | |
| Differentiation | | |
| Well | 4 (13.3%) | |
| Moderate | 13 (43.3%) | |
| Poor | 11 (36.7%) | |
| Unknown | 2 (6.7%) | |
| Clinical Stage | | |
| I | 10 (33.3%) | |
| II | 5 (16.7%) | |
| III | 11 (36.7%) | |
| IVA | 1 (3.3%) | |
| IVB | 3 (10%) | |
| Treatment | | |
| ESD | 3 (10%) | |
| Surgery | 15 (50%) | |
| Neoadjuvant therapy | 9 (30%) | |
| CRT | 2 (6.7%) | |
| CT | 1 (3.3%) | |
| Pathological Stage | | |
| IA | 5 (16.7%) | |
| IB | 6 (20%%) | |
| IIA | 1 (3.3%) | |
| IIB | 6 (20%) | |
| IIIA | 2 (6.7%) | |
| IIIB | 4 (13.3%) | |
| IVA | 2 (6.7%) | |
| IVB | 1 (3.3%) | |
| **C** | | |
| | ESCC patients (n = 18) | Control (n = 18) |
| Age (y) | 66 (38–81) | 64 (32–75) |
| Sex | | |
| Male | 11 (61.1%) | 10 (55.6%) |
| Female | 7 (38.9%) | 8 (44.4%) |
| Smoking | | |
| None | 3 (16.7%) | |
| Ex | 8 (44.4%) | |
| Current | 7 (38.9%) | |
| Tumor location | | |
| Ut | 4(22.2%) | |
| Mt | 9 (50%) | |
| Lt | 5 (27.8%) | |

(*Continued*)

**Table 1.** (Continued)

| | | |
|---|---|---|
| Ae | 0 | |
| Differentiation | | |
| Well | 2 (11.1%) | |
| Moderate | 9 (50%) | |
| Poor | 6 (33.3%) | |
| Unknown | 1 (5.6%) | |
| Clinical Stage | | |
| I | 7 (38.9%) | |
| II | 3 (16.7%) | |
| III | 5 (27.8%) | |
| IVA | 0 | |
| IVB | 3 (16.7%) | |
| Treatment | | |
| ESD | 4 (22.2%) | |
| Surgery | 4 (22.2%) | |
| Neoadjuvant therapy | 8 (44.4%) | |
| CRT | 0 | |
| CT | 2 (11.1%) | |
| Pathological Stage | | |
| IA | 6 (33.3%) | |
| IB | 2(11.1%) | |
| IIA | 1 (5.6%) | |
| IIB | 0 | |
| IIIA | 1 (5.6%) | |
| IIIB | 4 (22.2%) | |
| IVA | 0 | |
| IVB | 2 (11.1%) | |

CRT, definitive chemoradiotherapy; CT, palliative chemotherapy; ESD, endoscopic submucosal dissection; Ex, ex-smoker; Lt, lower thoracic; Mt, middle thoracic; Neo, neoadjuvant chemotherapy or chemoradiotherapy; S, surgery; Ut, upper thoracic.

deletions compared with mature miRs. To compare the read number of small RNAs between samples, total read numbers of each sample were normalized to 1,000,000 reads; in other words, each small RNA read number was calculated per 1,000,000.

Diagnostic biomarkers were identified by analyzing normalized read numbers of miR/iso-miRs between ESCC and HC using the Student t-test. As defined diagnostic biomarkers were identified in over 90% of samples of both the ESCC and HC groups, mean read numbers significantly differed more than 2-fold (p<0.05). Candidates miR/isomiRs which met our criteria for diagnostic biomarkers were entered stepwise into a multiple linear regression model to generate a diagnostic panel for ESCC. Minimum Bayesian Information Criteria (BIC) method was applied to select the best model. A panel index was calculated by assigning the read number of candidates of miR/isomiR selected by the diagnostic panel. Receiver operating characteristic (ROC) curves of the candidate of miR/isomiRs and panel index were generated to predict ESCC patients. The panel index was compared between patients with HGD, EAD, and ESCC using the Student t-test, and between pre- and post-treatment using the paired t-test. Data are presented as numbers (%) or as mean ± standard deviation in normally distributed

- Extraction miRNA/isoMIRs from serum samples
- Comparative analysis using small RNA sequencing

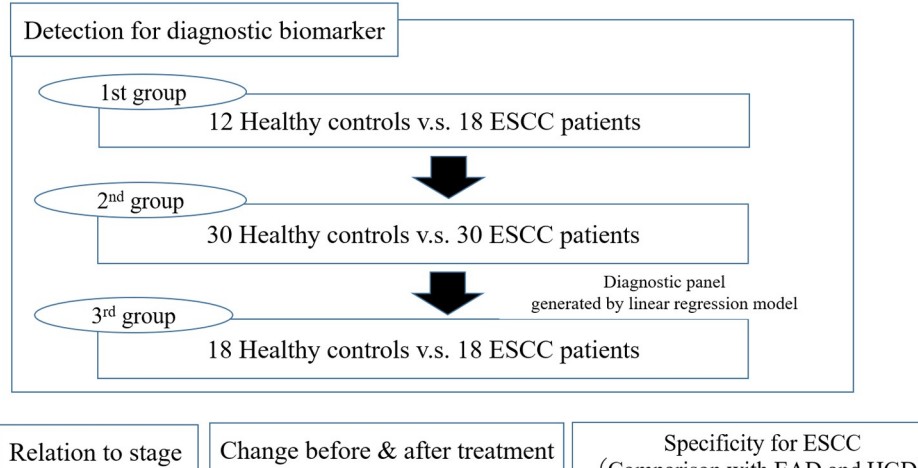

Fig 1. Overview of this study.

continuous variables. Frequencies were compared using the $\chi^2$ test for categorical variables and small samples were analyzed using Fisher's exact test.

All data were statistically analyzed using JMP® 14 (SAS Institute Inc., Cary, NC, USA).

# Results

## Identification of diagnostic biomarkers for ESCC

In the 1st group, 5451 miR/isomiRs were detected in at least one sample (S1 File). Among these, 303 miR/isomiRs were detected in over 90% of each group. Twenty-eight mature miRs and 60 isomiRs met the criteria for diagnostic biomarkers. These 88 candidates were validated in the 2nd group. The results of sequencing in the 2nd group were shown in S2 File. As a result, 9 mature miRs and 15 isomiRs also met the criteria in the 2nd group. Table 2 shows the profile of these candidates of miR/isomiRs, read number for ESCC and HC, fold change, and p-value in the 1st and 2nd group.

## Creation of the diagnostic panel

Twenty-four candidates which met the criteria for diagnostic biomarker were entered into a multiple regression model with stepwise selection to generate diagnostic panel for ESCC. The model entered variables to forward, and judged combination of three variables as optimal; one mature miRNA (miR-30a-5p) and two isomiRs [miR-574-3p (3' deletion A) and miR-205-5p (3' deletion G)] (S1 Fig and S3 File). Individual read numbers of miR/isomiRs used in the diagnostic panel are shown in Fig 2, and their diagnostic significance is shown in S2 Fig. The panel index was calculated from estimates indicated by the regression model [Panel Index = 0.83 +0.015 ×miR-574-3p(3' deletion A)+0.004×miR-205-5p(3' deletion G)+0.0018×miR-30a-5p]. This index was significantly higher in patients with ESCC than HC (3.1±1.3 vs. 13.3±8.9, p<0.001). The area under the receiver operating characteristics (ROC) curves (AUC) of the panel index used to predict ESCC patients was 0.95 (95% CI, 0.91–1.0, p<0.001; Fig 3). Using the optimal cut off value of 4.0, sensitivity and specificity was 93.8% and 81%, respectively (Fig 4A).

**Table 2. Profile of biomarker candidates of miR/isomiR.**

| Small RNA | L | Sequence | 1st group | | | | 2nd group | | | |
|---|---|---|---|---|---|---|---|---|---|---|
| | | | ESCC | HC | FC | P | ESCC | HC | FC | P |
| miR-885-5p | 22 | UCCAUUACACUACCCUGCCUCU | 4147±2974 | 169±153 | 24.4 | <0.001 | 1915±2301 | 119±138 | 16.1 | <0.001 |
| miR-574-3p (3' deletion A)* | 21 | CACGCUCAUGCACACACCCAC | 372±228 | 65±40 | 5.69 | <0.001 | 277±254 | 49±29 | 5.63 | <0.001 |
| miR-378a-3p | 22 | ACUGGACUUGGAGUCAGAAGGC | 1977±2389 | 454±229 | 4.36 | 0.042 | 1158±962 | 360±212 | 3.21 | <0.001 |
| miR-375-3p | 22 | UUUGUUCGUUCGGCUCGCGUGA | 598±472 | 95±71 | 6.3 | 0.001 | 618±811 | 97±82 | 6.4 | 0.001 |
| miR-365a/365b-3p | 22 | UAAUGCCCCUAAAAAUCCUUAU | 1062±804 | 185±167 | 5.73 | 0.001 | 1050±1187 | 310±238 | 3.38 | 0.001 |
| miR-335-5p (3' deletion U)* | 22 | UCAAGAGCAAUAACGAAAAAUG | 746±558 | 213±123 | 3.5 | 0.006 | 509±554 | 166±123 | 3.05 | 0.0015 |
| miR-205-5p (3' deletion G)* | 21 | UCCUUCAUUCCACCGGAGUCU | 814±508 | 152±142 | 5.36 | <0.001 | 1804±2567 | 187±177 | 9.62 | 0.0031 |
| miR-199a-1/a-2-3p (3' deletion A)* | 21 | ACAGUAGUCUGCACAUUGGUU | 1050±683 | 448±244 | 2.34 | 0.008 | 800±1162 | 362±225 | 2.2 | 0.002 |
| miR-193a-5p | 22 | UGGGUCUUUGCGGGCGAGAUGA | 2321±2301 | 752±450 | 3.09 | 0.019 | 1991±1162 | 702±395 | 2.72 | <0.001 |
| miR-148-3p (3' deletion U)* | 21 | UCAGUGCACUACAGAACUUUG | 2593±1352 | 804±239 | 3.22 | <0.001 | 1946±2027 | 950±306 | 2.05 | 0.011 |
| miR-145-5p (3' deletion U)* | 22 | GUCCAGUUUUCCCAGGAAUCCC | 12103±7220 | 1980±2218 | 6.11 | <0.001 | 8023±9544 | 1809±983 | 4.43 | <0.001 |
| miR-145-5p (3' deletion CU)* | 21 | GUCCAGUUUUCCCAGGAAUCC | 5551±3534 | 1467±1459 | 3.78 | 0.001 | 5990±5976 | 1955±1190 | 3.06 | <0.001 |
| miR-125b-1/b-2-5p (3' deletion GA)* | 20 | UCCCUGAGACCCUAACUUGU | 1677±1448 | 211±152 | 7.92 | <0.001 | 1133±1295 | 184±133 | 6.13 | <0.001 |
| miR122-5p | 22 | UGGAGUGUGACAAUGGUGUUUG | 15588±35076 | 1228±963 | 12.7 | 0.04 | 5694±22830 | 622±437 | 9.15 | <0.001 |
| miR122-5p (3' deletion G)* | 21 | UGGAGUGUGACAAUGGUGUUU | 35663±35332 | 2376±1296 | 15 | <0.001 | 17822±22707 | 2024±1671 | 8.8 | <0.001 |
| miR122-5p (3' deletion UG)* | 20 | UGGAGUGUGACAAUGGUGUU | 1373±1488 | 164±122 | 8.3 | 0.011 | 538±596 | 123±141 | 4.34 | <0.001 |
| miR-99a-5p (3' deletion G)* | 21 | AACCCGUAGAUCCGAUCUUGU | 4521±3794 | 327±358 | 13.8 | 0.001 | 363±362 | 152±146 | 2.38 | 0.003 |
| miR-34a-5p | 22 | UGGCAGUGUCUUAGCUGGUUGU | 927±713 | 131±95 | 6.97 | 0.001 | 385±229 | 115±77 | 3.18 | <0.001 |
| miR-30a-5p | 22 | UGUAAACAUCCUCGACUGGAAG | 1514±824 | 334±159 | 4.53 | <0.001 | 1563±1034 | 451±377 | 3.46 | <0.001 |
| miR-27b-3p (3' deletion C)* | 20 | UUCACAGUGGCUAAGUUCUG | 701±491 | 191±99 | 3.67 | 0.002 | 640±625 | 178±89 | 3.56 | <0.001 |
| miR-22-3p | 22 | AAGCUGCCAGUUGAAGAACUGU | 6665±3375 | 3347±1917 | 2 | 0.006 | 5497±2945 | 2707±1126 | 2.03 | <0.001 |
| miR-10b-5p (3' deletion G)* | 22 | UACCCUGUAGAACCGAAUUUGU | 1842±1069 | 648±296 | 2.99 | 0.003 | 1572±951 | 562±344 | 2.8 | <0.001 |
| miR-10b-5p (3' deletion GU)* | 21 | UACCCUGUAGAACCGAAUUUG | 422±290 | 140±71 | 2.84 | 0.001 | 524±308 | 214±142 | 2.45 | <0.001 |
| miR-10a-5p (5'deletion U, 3'deletion G)* | 21 | ACCCUGUAGAUCCGAAUUUGU | 238±45 | 52±14 | 4.54 | 0.001 | 413±167 | 149±42 | 2.77 | 0.004 |

*, isomiR; ESCC, patients with esophageal squamous cell carcinoma; FC, fold change, HC, healthy control; L, length of sequence; miR, micro RNA.

## Validating the diagnostic panel

To confirm the diagnostic value of our panel for ESCC, we tested it in another independent group (3rd group S4 File). Mean value of the panel index was 16.8±20.8 and 3.6±1.3 in patients with ESCC and HC, respectively (p<0.001). Diagnostic sensitivity and specificity using same cut off value was 88.9% and 72.3% (Fig 4B). AUC of the ROC curve was 0.89 (95%CI, 0.78–1.0, p<0.001; S3 Fig).

## Comparison of panel index between patients with ESCC, EAD and HGD

The profiles of miR/isomiRs were also investigated in patients with EAD and HGD (S5 File). The mean panel index of patients with EAD and HGD was 4.2±1.7 and 6.2±4.5, respectively. These values were significantly lower than that of patients with ESCC. In contrast, while they were also higher than in HC, the difference was not statistically significant (Fig 5).

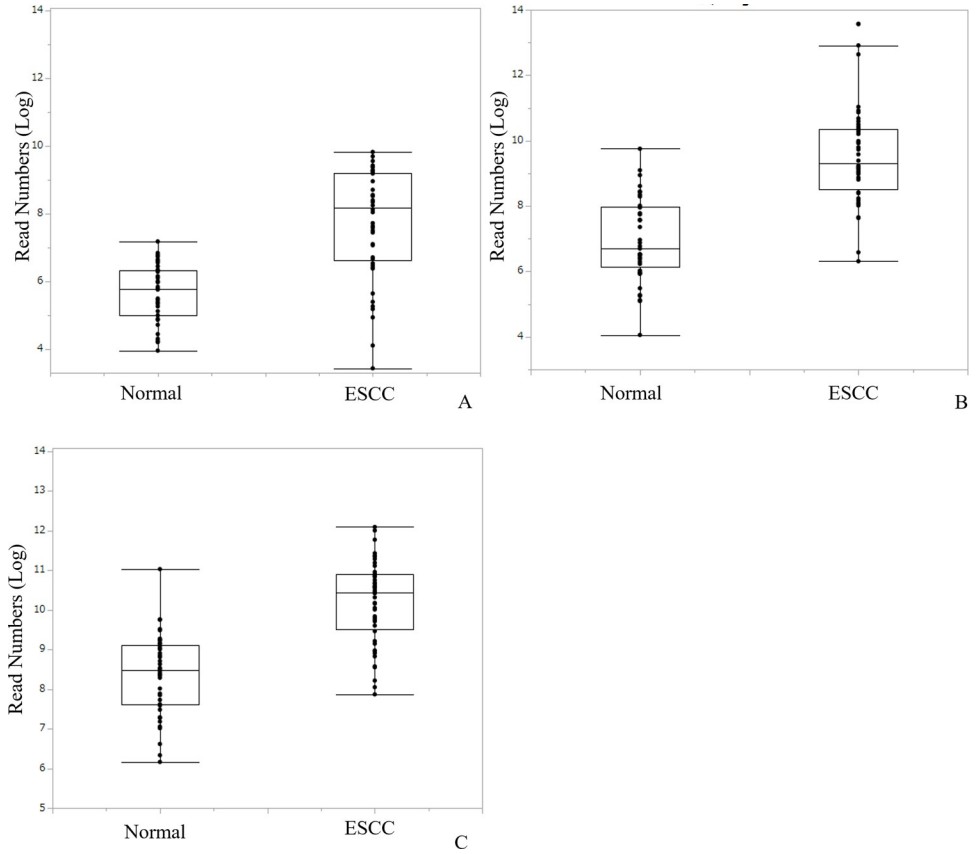

**Fig 2. Read numbers of miR/isomiRs applied to the diagnostic panel in healthy control and patients with ESCC.**
Boxplot of read numbers of miR/isomiRs applied to the diagnostic panel in healthy control and patients with ESCC;
miR-574-3p (3' deletion A) (A), miR-205-5p (3' deletion G) (B), and miR-30a-5p (C).

### Relationship between panel index and clinical and pathological stage

Fig 6A shows the panel index of patients with ESCC according to clinical stage. Mean panel index of patients with stage I, II, III, and IV disease was 11.4±6.3, 13.8±7.2, 12.8±11.7 and 31.2 ±28.9, respectively. Patients with stage IV tend to have a higher index compared with those with stage I–III disease, but the difference was not significant. A similar trend was seen by pathological stage (Fig 6B). While patients with clinical stage I disease tended to have a lower index than those with advanced stage disease, the index was still significantly higher than that in HCs. Diagnostic sensitivity and specificity using cut off value of 4.0 was 91.0% and 77.4%, respectively. AUC of the ROC curve was 0.93 (95%CI, 0.85–1.0, p<0.001; S4 Fig).

### Time course of change in panel index of patients with ESCC during treatment and at recurrence

The 22 paired samples at pre- and post- treatment were investigated for the expression of miR/ isomiR, and a panel index was calculated. Mean panel index after treatment was significantly decreased compared with that before treatment (6.2±5.6 vs 11.6±11.5, p = 0.03; Fig 7) Eighteen cases (81.8%) showed a decrease in panel index after treatment compared with before. Mean decreased ratio was 0.28±0.15 (S5 Fig). Fig 8 shows the time course of panel index changes in the four patients who experienced postoperative recurrence. Panel index of all four patients

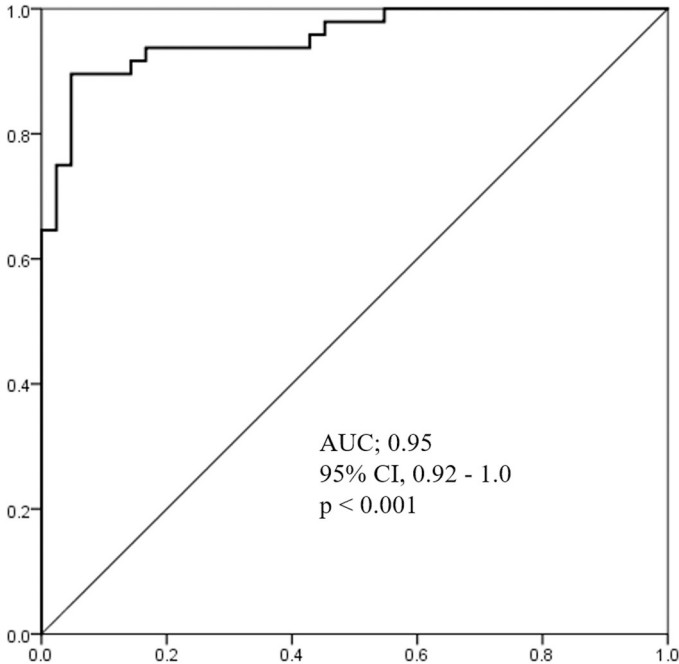

**Fig 3. Receiver operating characteristics (ROC) curves of the panel index in the 1st and 2nd groups.** Area under receiver operating characteristics curves (AUC) for panel index to predict esophageal squamous cell carcinoma: AUC, 0.95; 95% CI, 0.92–1.0; p<0.001.

decreased after treatment compared with those before, and increased again at recurrence in three (S6 Fig).

## Discussion

We aimed to identify the clinical significance of circulating miR/isomiRs in patients with ESCC detected by NGS. We identified 24 miR/isomiRs as diagnostic biomarkers by comparison between ESCC patients and HCs in different two cohorts. The diagnostic panel generated by these candidates had high accuracy in the diagnosis of ESCC.

Early detection is important in improving outcomes in patients with ESCC. Endoscopic screening is the standard for detecting superficial ESCC [23]. Although recent advances in diagnostic technology for cancer such as narrow band imaging provide high accuracy, the relatively low incidence of ESCC renders population-based screening ineffective. Endoscopy also causes chest discomfort in all subjects and sometimes has unpleasant adverse effects, such as aspiration pneumonia. Accordingly, screening for ESCC should be limited to individuals at high risk. In fact, screening endoscopy has been proven effective in detecting early-stage ESCC and precancerous lesions in a high-risk region in China [24]. However, regional differences in the occurrence of esophageal cancer are not seen in Japan or Western countries, indicating the need for biomarkers that can detect patients with ESCC. Given the low invasiveness of blood sampling, circulating small RNA might be an ideal biomarker candidate. Indeed, many studies have confirmed the usefulness of circulating miR in detecting cancer. Theoretically, isomiRs might also be powerful biomarkers, like mature miRs. However, few studies have examined this possibility, primary because the similarity in the sequences of isomiR and mature miR makes it technically difficult to distinguish them by usual quantitative polymerase chain reaction (qPCR). Recent developments in deep sequencing systems, represented by NGS, allow the

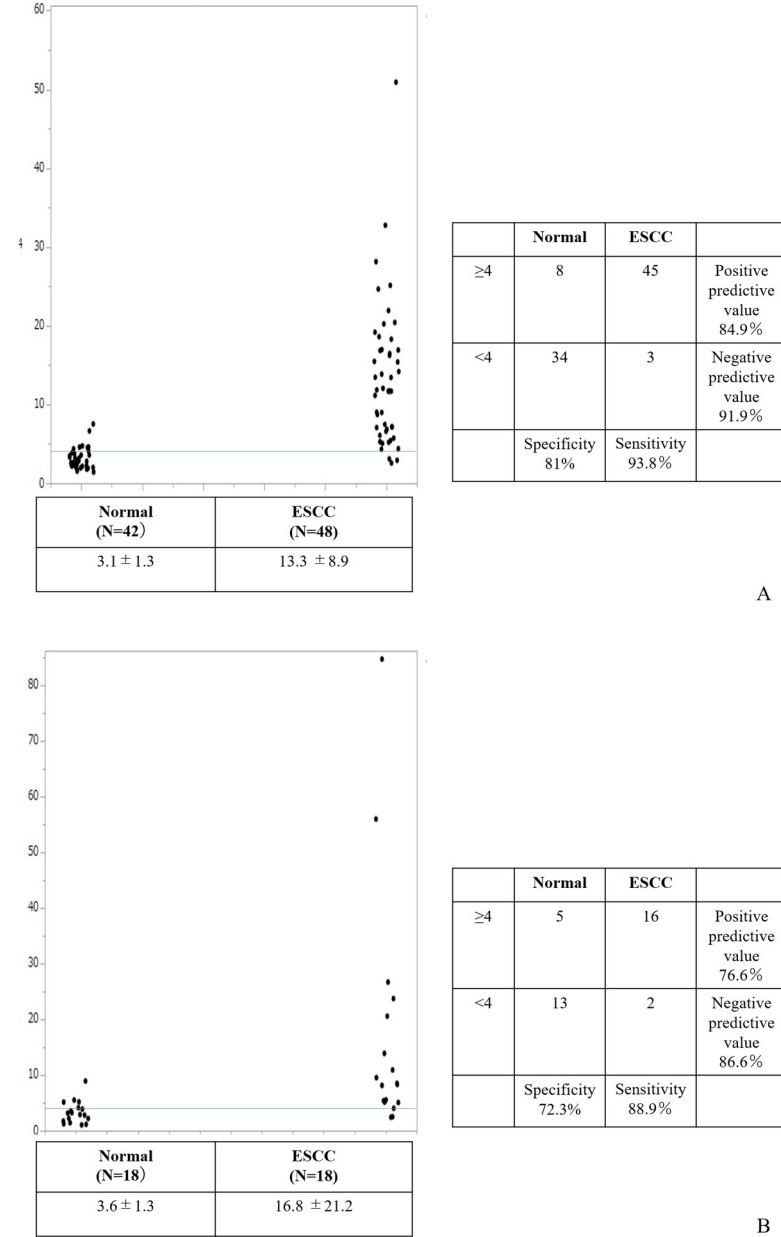

**Fig 4. Significance of the diagnostic panel.** Mean panel index was significantly higher in patients with ESCC than HC (13.3±8.9 vs. 3.1±1.3, p<0.001) in 1st and 2nd groups. Diagnostic sensitivity and specificity were 93.8% and 81%, using cut off value for the panel index of 4.0 in the 1st and 2nd groups (A). Mean panel index was significantly higher in patients with ESCC than HC (16.8±21.2 vs. 3.6±1.3, p<0.001) in the 3rd group. Diagnostic sensitivity and specificity were 88.9% and 72.2%, using a cut off value for the panel index of 4.0 in the 3rd group (B).

detection of even slight differences in small RNAs and the identification of isomiRs. Several researchers have described studies focused on isomiRs from tumors. Wu et al reported that expression of isomiRs in colorectal tissue differed between normal mucosa, adenoma, and adenocarcinoma [25]. Roberts et al reported that circulating small RNA, including isomiR, were associated with colorectal adenoma [26]; and Mjelle et al identified circulating miR/isomiR associated with metastasis of rectal cancer [27]. However, few studies have examined differences in miR/isomiR between cancer patients and healthy individuals. To our knowledge, our

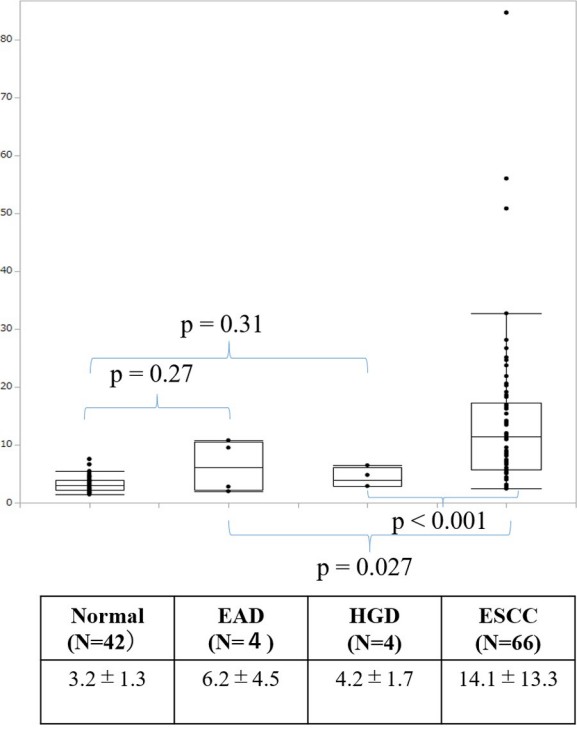

| Normal (N=42) | EAD (N=4) | HGD (N=4) | ESCC (N=66) |
|---|---|---|---|
| 3.2 ± 1.3 | 6.2 ± 4.5 | 4.2 ± 1.7 | 14.1 ± 13.3 |

**Fig 5. Comparison of diagnostic panel index in patients with esophageal dysplasia, adenocarcinoma and squamous cell carcinoma.** Mean panel indices of patients with esophageal adenocarcinoma (4.2±1.7) and high-grade dysplasia (6.2±4.5) were significantly lower than patients with ESCC (14.1±13.3), but did not significantly defer compared from the HC (3.2 ± 1.3).

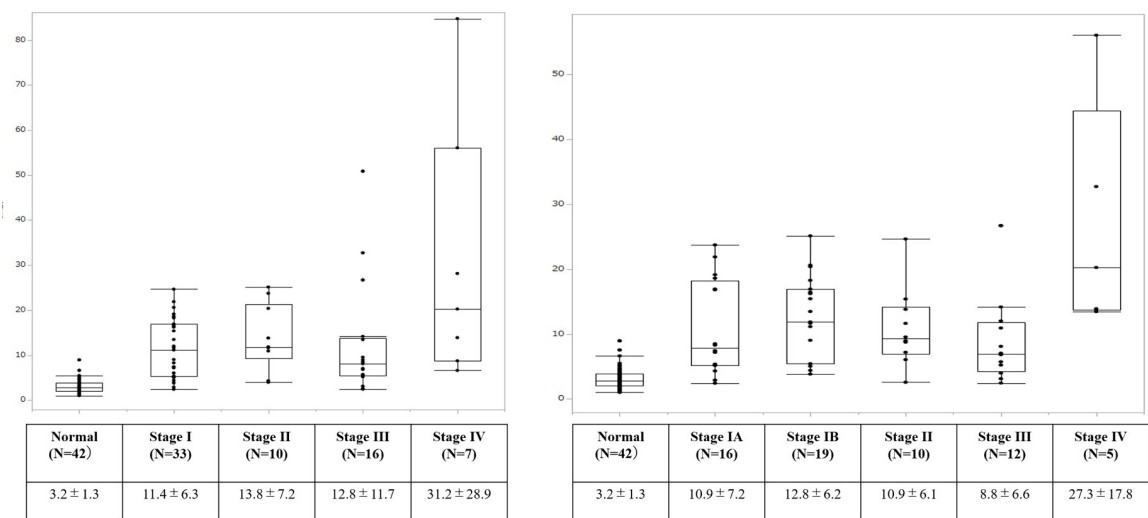

| Normal (N=42) | Stage I (N=33) | Stage II (N=10) | Stage III (N=16) | Stage IV (N=7) |
|---|---|---|---|---|
| 3.2 ± 1.3 | 11.4 ± 6.3 | 13.8 ± 7.2 | 12.8 ± 11.7 | 31.2 ± 28.9 |

| Normal (N=42) | Stage IA (N=16) | Stage IB (N=19) | Stage II (N=10) | Stage III (N=12) | Stage IV (N=5) |
|---|---|---|---|---|---|
| 3.2 ± 1.3 | 10.9 ± 7.2 | 12.8 ± 6.2 | 10.9 ± 6.1 | 8.8 ± 6.6 | 27.3 ± 17.8 |

**Fig 6. Comparison of diagnostic panel index according to stage.** Patients of all stages had a significantly higher panel index than the healthy control. Panel index of patients with clinical stage IV disease (31.2±28.9) tended to be high compared with clinical stages I (11.4 ±6.3), II (13.8±7.2), and III (12.8±11.7), although without statistical significance (A). Panel index of patients with pathological stage IV disease (27.3±17.8) tended to be high compared with pathological stage IA (10.9±7.2), IB (12.8±6.2), II (10.9± 6.1), and III (8.8 ±6.6), although without statistical significance (B).

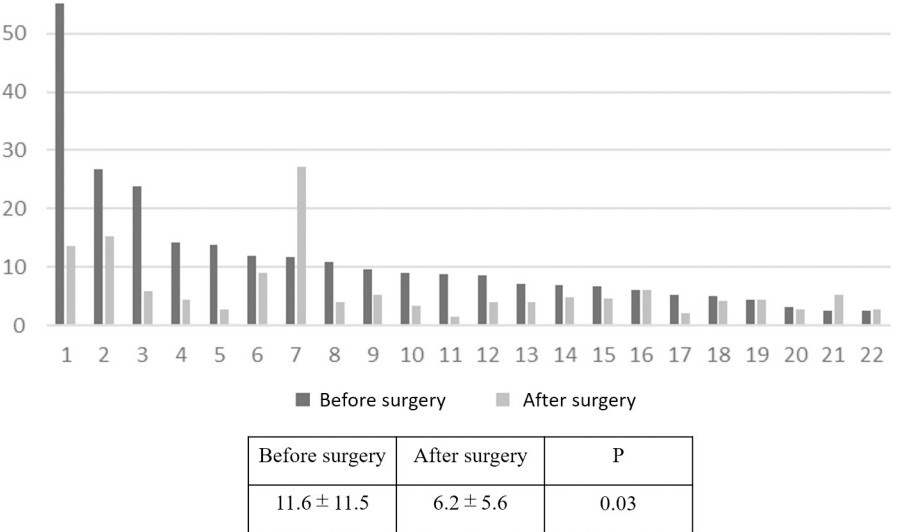

| Before surgery | After surgery | P |
|---|---|---|
| 11.6 ± 11.5 | 6.2 ± 5.6 | 0.03 |

**Fig 7. Comparison of panel index before and after treatment.** Mean panel index of after treatment was significantly decreased compared with before treatment (6.2±5.6 vs 11.6±11.5, p = 0.03).

present study is the first to show the usefulness of combination of circulating miR/somiRs detected by NGS in the diagnosis of esophageal cancer.

Our diagnostic panel was generated by comparing patients with ESCC at all stages and HC controls. The panel was useful in detecting patients even at stage I, and in distinguishing patients with ESCC from those with from HGD and EAD. These findings would also be useful in distinguishing individuals at high risk of ESCC but without significant symptoms, and in population-level endoscopy screening.

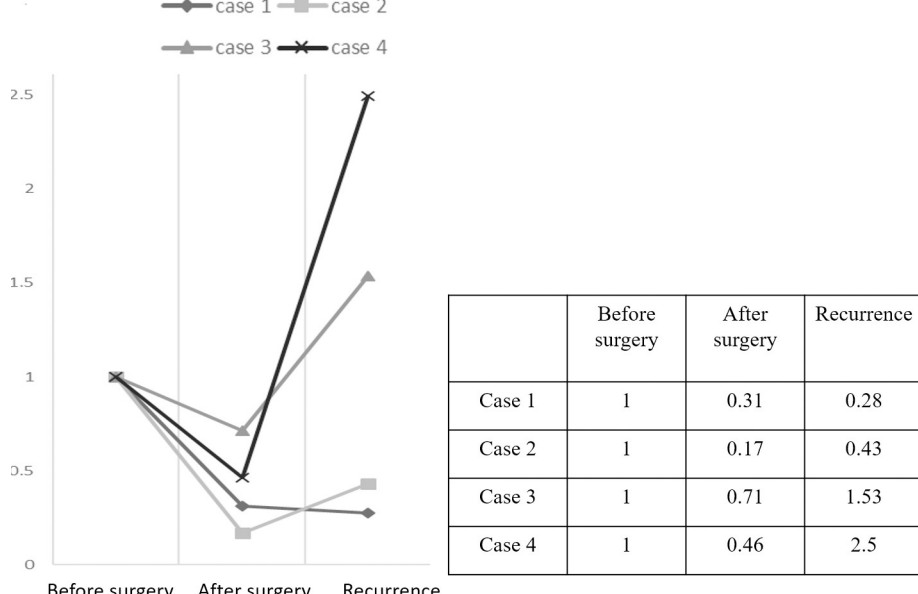

| | Before surgery | After surgery | Recurrence |
|---|---|---|---|
| Case 1 | 1 | 0.31 | 0.28 |
| Case 2 | 1 | 0.17 | 0.43 |
| Case 3 | 1 | 0.71 | 1.53 |
| Case 4 | 1 | 0.46 | 2.5 |

**Fig 8. Time course of panel index change ratio in patients who experienced postoperative recurrence.** Change rate of the panel index when pre-treatment panel index is 1.0. Index decreased in all four patients after treatment compared with before, re-increased in three of four who experienced recurrence.

This panel includes one mature miR and two isomiRs. According to previous reports, miR-30a-5p plays a dual role in different types of cancer as either an oncogene or onco-suppressor [28]. Function of miR-30a-5p as cancer activators has been reported in pharyngeal cancer [29], ovarian cancer [30] and glioma [31]. Their expression profiles also differ between cancer and normal tissue. Kimura et al reported that miR-30a-5p is up-regulated in ESCC, as well as in a head and neck squamous cell carcinoma cell line compared with normal squamous epithelial cell lines [32]. In contrast, circulating miR-30a-5p is down-regulated in patients with EAD compared with healthy control [33]. MiR-205-5p also has several functions which appear to depend on cellular context and tumor subtype. It is also reported to have specific features in squamous cell carcinoma, and is a reliable biomarker to distinguish squamous cell carcinoma from other subtypes in non-small lung cell cancer tissue [34–36]. Circulating miR-205-5p is up-regulated in patients with lung squamous cell carcinoma [36] and cervical cancer [37]. Moreover, a recent study found that miR-205-5p has different function in squamous cell carcinoma and adenocarcinoma in the esophagus [38]. MiR-574-3p is upregulated in hepatocellular carcinoma [39] and prostate cancer [40], and is positively associated with the proliferation of osteosarcoma [41]. Moreover, Krishnan et al described the prognostic impact of miR-574-3p detected by NGS from breast cancer tissue [42].

Of note, these previous reports dealt with the mature miR-205-5p and miR-574-3p whereas our diagnostic panel included isomiR. The two types were previously thought to have a similar function because of their similar sequence, but more recent studies have identified that they have different functions [19, 43, 44]. In fact, the target messenger RNA of isomiR has concordance and discordance with mature miR, in accordance with the difference between them in sequence [45]. Further study is therefore needed to identify whether these isomiRs have the same function as mature miRs.

Although our panel is not aimed at detecting postoperative recurrence, the panel index was decreased after treatment compared with that before treatment in almost all cases, and re-increased at recurrence in three of four patients. Some miR/isomiRs likely change as a reflection of tumor volume. Supporting this, Komatsu et al reported that levels of circulating miR-25 changed before and after surgery [14]. Follow-up of certain miR/isomiRs by post-treatment survey might be worthwhile.

Several limitations of our study warrant mention. Because few studies have dealt with circulating miR/isomiR detected by NGS, no clear consensus exists for the normalization of miR/isomiR, nor is there a consistent method for analyzing data. We normalized read number as 1,000,000 reads per sample in accordance with a previous report. If normalization and data analysis methods change, different results will be obtained. Our results were also influenced by the number of samples assigned to each group and the method of statistical analysis. Obtaining repeatable results in future studies therefore requires establishment of a concrete consensus. External validation is preferred to confirm accuracy of our results, but it is difficult because there is no public database containing information on circulating isomiR in ESCC patients. Therefore we tested the application for our diagnostic panel using another cohort, but it was using retrospective single institution samples after all. Prospective confirmation study is needed before clinical application. We investigated miR/isomiR profiles from serum samples stored for several periods. Although there was no substantial difference between the retention periods of samples from patients and HC, the possibility that this difference affected the results cannot be denied. It remains unclear whether these candidate miR/isomiRs for diagnostic biomarkers differ between normal squamous epithelium and squamous cell carcinoma tissue, as does the function of these candidates in vivo, and further studies are needed to clarify these questions. We focused on miR/isomiR in the present study, but other small RNAs are

abundant in tissue and blood and can be detected by NGS. These small RNAs might include other powerful biomarkers of ESCC.

## Conclusion

We focused on circulating miR/isomiR detected by NGS as novel biomarkers of ESCC. Our diagnostic panel had high accuracy in diagnosis and high specificity as a biomarker of ESCC. Although a number of problems must be resolved before clinical application, miR/isomiRs detected by NGS could serve as novel biomarkers of ESCC.

## Supporting information

**S1 Fig. Bayesian information criteria (BIC) value according to combination of variables.** The forward stepwise model showed the combination of miR-574-3p (3' deletion A), miR-205-5p (3' deletion G), and miR-30a-5p indicated the minimum BIC.
(TIF)

**S2 Fig. Diagnostic significance of miR/isomiRs used in the diagnostic panel.** Area under the receiver operating characteristics curves (AUC) for miR-574-3p (3' deletion A) (A), miR-205-5p (3' deletion G) (B), and miR-30a-5p (C) to predict esophageal squamous cell carcinoma. miR-574-3p (3' deletion A): AUC, 0.84; 95% CI, 0.75–0.93; p<0.001; miR-205-5p (3' deletion G): AUC, 0.92; 95% CI, 0.86–0.97; p<0.001, and miR-30a-5p: AUC, 0.89; 95% CI, 0.82–0.96; p<0.001.
(TIF)

**S3 Fig. Receiver operating characteristics (ROC) curves of the panel index in the 3rd group.** Area under the receiver operating characteristics curves (AUC) for the panel index to predict esophageal squamous cell carcinoma: AUC, 0.89; 95% CI, 0.78–1.0; p<0.001.
(TIF)

**S4 Fig. Significance of the diagnostic panel for clinical stage I ESCC.** Area under receiver operating characteristics curves (AUC) for panel index to predict stage I esophageal squamous cell carcinoma: AUC, 0.93; 95% CI, 0.9–1.0; p<0.001. Diagnostic sensitivity and specificity were 90.4% and 78.4%, using a cut off value for the panel index of 4.0.
(TIF)

**S5 Fig. Panel index change ratio after treatment compared with before treatment.** Change rate of the panel index when the pre-treatment panel index is 1.0. Mean post-treatment panel index was significantly decreased compared with pre-treatment (mean decrease in ratio was 0.28±0.15).
(TIF)

**S6 Fig. Comparison of panel index before, and after treatment, and at recurrence.** Time course of changes in panel index in patients who experienced post-operative recurrence.
(TIF)

**S1 File. Expression profile of miR/isomiR in the 1st group.**
(XLSX)

**S2 File. Expression profile of miR/isomiR in the 2nd group.**
(XLSX)

**S3 File. Stepwise regression model to generate a diagnostic panel for ESCC.**
(XLSX)

**S4 File. Expression profile of miR/isomiR in the 3$^{rd}$ group.**
(XLSX)

**S5 File. Expression profile of miR/isomiR in patients with esophageal dysplasia and adenocarcinoma.**
(XLSX)

## Acknowledgments

The authors thanks Prof. Junko Tanaka (Professor of Department of Epidemiology, Infectious Disease Control and Prevention, Hiroshima university) for supervising statistical analysis.

## Author Contributions

**Conceptualization:** Yasuhiro Tsutani, Morihito Okada, Hidetoshi Tahara.

**Data curation:** Yuta Ibuki, Yukie Nishiyama, Manabu Emi.

**Formal analysis:** Yuta Ibuki.

**Funding acquisition:** Morihito Okada, Hidetoshi Tahara.

**Investigation:** Yuta Ibuki, Manabu Emi.

**Methodology:** Yukie Nishiyama.

**Resources:** Manabu Emi, Yoichi Hamai, Hidetoshi Tahara.

**Supervision:** Yukie Nishiyama, Yasuhiro Tsutani, Yoichi Hamai, Morihito Okada, Hidetoshi Tahara.

**Validation:** Yukie Nishiyama.

**Visualization:** Yuta Ibuki.

**Writing – original draft:** Yuta Ibuki.

**Writing – review & editing:** Yasuhiro Tsutani, Morihito Okada, Hidetoshi Tahara.

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
