## [Decision Letter · Decision Letter 0]

30 Dec 2019

PONE-D-19-31398

Circulating microRNA/isomiRs as novel biomarkers of esophageal squamous cell carcinoma.

PLOS ONE

Dear Prof Tahara,

Thank you for submitting your manuscript to PLOS ONE. After careful consideration, we feel that it has merit but does not fully meet PLOS ONE’s publication criteria as it currently stands. Therefore, we invite you to submit a revised version of the manuscript that addresses the points raised during the review process.

This is a retrospective, single-institute study that examines microRNAs (miRNAs) and isomicroRNAs (isomiRs) using the next generation sequencer (NGS) in serum samples from 66 patients with esophageal squamous cell carcinoma (ESCC) designed to identify circulating miR/isomiRs that could be used as novel biomarker for ESCC. You have demonstrated that your diagnostic panel index derived from the regression analysis with miR-30a-5p and isoforms of miR-574-3p and miR-205-5p had high accuracy in the diagnosis of ESCC. You have suggested that MiR/isomiRs detected by NGS can be novel biomarkers of ESCC. 

You have demonstrated that twenty-four miRNA/isomiR candidates are entered into a multiple regression model with stepwise selection to generate diagnostic panel for ESCC. Despite your appreciable efforts with NGS, major limitation of your study is the very small number of discovery cohort to perform regression analysis. In regression analysis, 10 observations per variable are recommended. So, generation of panel index with coefficients from stepwise regression analysis is not statistically appropriate. You would be better to combine your cohorts to maximize the number of samples, which approach would be more statistically reasonable. As the independent validation of your findings, you can use publicly available ESCC cohorts from GEO and TCGA database.  

Normally distributed continuous variables are reported as mean ± standard deviation; non–normally distributed continuous variables, as median and range (or IQR).

Many grammar and typo errors are found. Thorough editing is required.

We would appreciate receiving your revised manuscript by Feb 10 2020 11:59PM. To enhance the reproducibility of your results, we recommend that if applicable you deposit your laboratory protocols in protocols.io, where a protocol can be assigned its own identifier (DOI) such that it can be cited independently in the future. For instructions see: http://journals.plos.org/plosone/s/submission-guidelines#loc-laboratory-protocols

We look forward to receiving your revised manuscript.

Kind regards,

Hyun-Sung Lee, M.D., Ph.D.

Academic Editor

PLOS ONE

Journal Requirements:

"Prof. Hidetoshi Tahara is representative director of a university-originated venture, MiRTeL Co. Ltd., which provides commercial microRNA panel services. The remaining authors declare no potential conflict of interest."

We note that one or more of the authors are employed by a commercial company: MiRTeL Co. Ltd..

1.) Please provide an amended Funding Statement declaring this commercial affiliation, as well as a statement regarding the Role of Funders in your study. If the funding organization did not play a role in the study design, data collection and analysis, decision to publish, or preparation of the manuscript and only provided financial support in the form of authors' salaries and/or research materials, please review your statements relating to the author contributions, and ensure you have specifically and accurately indicated the role(s) that these authors had in your study. You can update author roles in the Author Contributions section of the online submission form.

2.) Please also provide an updated Competing Interests Statement declaring this commercial affiliation along with any other relevant declarations relating to employment, consultancy, patents, products in development, or marketed products, etc. 

Reviewers' comments:

Reviewer's Responses to Questions

**Comments to the Author**

1. Is the manuscript technically sound, and do the data support the conclusions?

Reviewer #1: Yes

Reviewer #2: Yes

2. Has the statistical analysis been performed appropriately and rigorously? 

Reviewer #1: Yes

Reviewer #2: Yes

3. Have the authors made all data underlying the findings in their manuscript fully available?

Reviewer #1: Yes

Reviewer #2: Yes

4. Is the manuscript presented in an intelligible fashion and written in standard English?

Reviewer #1: No

Reviewer #2: Yes

5. Review Comments to the Author

Reviewer #1: Some comments:

1. There are some english errors.

2. Spell out some words

3. Background in the abstract should be write in better way

4. Add the below references for the first sentence of "Introduction section"

a. Int J Cancer. 2019 Mar 15;144(6):1215-1226.

b. J Cell Physiol. 2018 Nov;233(11):8538-8550

5. Add the below references for this sentence:

"MicroRNA(miR)s are classified as small noncoding RNAs (19 - 25 nucleotides) which

regulate the expression of plural numbers of messenger RNAs"

a. J Cell Biochem. 2017 Dec;118(12):4121-4128

b. Adv Clin Chem. 2017;82:47-70.

c. J Cell Physiol. 2018 Feb;233(2):866-879.

Reviewer #2: The study tried to explore the biomarkers of ESCC from the microRNA/isomiRs in circulating blood based on the technique of NGS. It is a very interesting and important study; the research was well conducted and the manuscript is well written and easy to follow. There are some specific suggestions for the revision of the paper according bellow that should help to improve the readability of the paper for the PLOS ONE audience.

1. In this nested case-control study, the authors select almost equal number (18 vs 12, 30 vs 30, 18 vs 18) of HC individuals as the patients after the age, sex matching. As I see, the result of the first group was the most important, why the number of HC was less than the case group? How did the authors select the HC individuals? If possible, a larger sample size may been more convinced.

2. In the 3rd group, the authors tried to test the “biomarkers” found from the two groups. Actually, to test the value of your discoveries, It seemly better to conduct a pure prospective case-control study instead of the ESCC patients only. How did the authors think?

3. The authors should check the spellings and grammar. Such as the “sere” in line 81.

6. PLOS authors have the option to publish the peer review history of their article (what does this mean?). If published, this will include your full peer review and any attached files.

Reviewer #1: No

Reviewer #2: No

---

## [Author Response · Author response to Decision Letter 0]

12 Feb 2020

Responses to the comments of Editor

We gratefully appreciate your review our manuscript and the helpful suggestions. Our point-by-point responses to your comments and suggestions are listed below.

Comment 1

Despite your appreciable efforts with NGS, major limitation of your study is the very small number of discovery cohort to perform regression analysis. In regression analysis, 10 observations per variable are recommended. So, generation of panel index with coefficients from stepwise regression analysis is not statistically appropriate. You would be better to combine your cohorts to maximize the number of samples, which approach would be more statistically reasonable. As the independent validation of your findings, you can use publicly available ESCC cohorts from GEO and TCGA database.

Response

As suggested by the editor, limited number of samples was limitation of this study. NGS analysis is difficult to analyze many samples due to cost for now. We analyzed more than 100 samples of patients and control to discover biomarker of ESCC. This number is the largest effort ever reported included other cancer.

It is well known that different RNA extraction methods strongly affected the NGS results, and there is no public database containing information on circulating isomiR in ESCC patients. Therefore, external validation is difficult, and internal validation was proceeded by dividing samples in this study. Thus, prospective study is needed before clinical application. Therefore, we added this sentence in the documents (‘‘We tested the application for our diagnostic panel using another cohort, but it was using retrospective single institution samples after all. Therefore, prospective confirmation study is needed before clinical application.’’; line279-281). 

The editor also points out about stepwise procedure. We understand there is a recommendation the number of covariates should be "10-15 observations per variable" as a "guide" for multiple regression. And we also understand there is criticism against stepwise method because there is unreproducible of variable selection and the methods select covariates without considering the clinical evaluation.

However, the "guide" is just reference criteria, and the suitable number of covariates will change depend on the variance and covariance of variables. Stepwise procedure is effective method to decide the suitable combination of covariates with considering the possible number of covariates and correlation between covariates, especially multicollinearity. Moreover, stepwise procedure has potential to find undiscovered confounding factors. 

Therefore, we applied the covariates both of clinical important covariate and variables selected by stepwise procedure We proceeded with this analysis in consultation with a clinical statistician (Prof. Junko Tanaka; Professor of Department of Epidemiology, Infectious Disease Control and Prevention, Hiroshima university).

Comment 2

Normally distributed continuous variables are reported as mean ± standard deviation; non–normally distributed continuous variables, as median and range (or IQR).

Response

As mentioned by the editor, we changed representation of normally distributed continuous variables as mean ± standard deviation. Specifically, we changed the representation of mean read number of patients and healthy control in Table 2.

Responses to the comments of Reviewer 1

We gratefully appreciate your review our manuscript and the helpful suggestions. Our point-by-point responses to your comments and suggestions are listed below.

Comment 1, 2

1. There are some english errors.

2. Spell out some words

Response

As suggested by the reviewer, we reviewed the text and corrected some errors (ex. ‘‘sere →were’’ in line 79). Abbreviations of micro RNA were not unified, so they were unified as ‘‘miR’’. We have also unified the description of the supporting information in the text (ex. S1 Fig. and S1 File). 

Comment 3 

Background in the abstract should be write in better way

Response

As suggested by reviewer, we changed background in the abstract as following “ MicroRNA (miR)s are promising diagnostic biomarkers of cancer. Recent next generation sequencer (NGS) studies have found that isoforms of micro RNA (isomiR) circulate in the bloodstream similarly to mature miRs. We hypothesized that combination of circulating miR and isomiRs detected by NGS are potentially powerful cancer biomarker. The present study aimed to investigate their application in esophageal cancer. ”

Comment 4, 5 

4. Add the below references for the first sentence of "Introduction section"

a. Int J Cancer. 2019 Mar 15;144(6):1215-1226.

b. J Cell Physiol. 2018 Nov;233(11):8538-8550

5. Add the below references for this sentence:

"MicroRNA(miR)s are classified as small noncoding RNAs (19 - 25 nucleotides) which regulate the expression of plural numbers of messenger RNAs"

a. J Cell Biochem. 2017 Dec;118(12):4121-4128

b. Adv Clin Chem. 2017;82:47-70.

c. J Cell Physiol. 2018 Feb;233(2):866-879. 

Response

As suggested by the reviewer, we added these literatures in References part.

Responses to the comments of Reviewer 2

We gratefully appreciate your review our manuscript and the helpful suggestions. Our point-by-point responses to your comments and suggestions are listed below.

Comment 1

In this nested case-control study, the authors select almost equal number (18 vs 12, 30 vs 30, 18 vs 18) of HC individuals as the patients after the age, sex matching. As I see, the result of the first group was the most important, why the number of HC was less than the case group? How did the authors select the HC individuals? If possible, a larger sample size may been more convinced. 

 Response

One of the main reasons is that time duration of blood collection was the same between patients and controls. As the reviewers mentioned, the first group result was important. We tried to get many candidates and validate them with many patients and controls to get precise diagnostic biomarkers. In our comparative analysis method, a smaller number of samples would yield more candidates. For this reason, more numbers of samples were allocated to the second group than to the first group.

We proceeded with this analysis in consultation with a clinical statistician (Junko Tanaka; Professor of Department of Epidemiology, Infectious Disease Control and Prevention, Hiroshima university).

Comment 2

In the 3rd group, the authors tried to test the “biomarkers” found from the two groups. Actually, to test the value of your discoveries, It seemly better to conduct a pure prospective case-control study instead of the ESCC patients only. How did the authors think?.

Response 

We think that prospective case-control testing is necessary for clinical use as pointed out by the reviewers. We did internal validation instead of prospective study. It is the limitations of this study. We will be conducting a prospective study based on results of the current study.

We added following the following statements to line 270-272 ‘‘We tested the application for our diagnostic panel using another cohort, but it was using retrospective single institution samples after all. Therefore, prospective confirmation study is needed before clinical application.’’

 Comment 3

 The authors should check the spellings and grammar. Such as the “sere” in line 81. 

Response

As suggested by the reviewer, we reviewed the text and corrected some errors including in line 79. Abbreviations of micro RNA were not unified, so they were unified as ‘‘miR’’. We have also unified the description of the supporting information in the text (ex. S1 Fig. and S1 File).

---

## [Decision Letter · Decision Letter 1]

3 Mar 2020

PONE-D-19-31398R1

Circulating microRNA/isomiRs as novel biomarkers of esophageal squamous cell carcinoma.

PLOS ONE

Dear Prof Tahara,

Thank you for submitting your manuscript to PLOS ONE. After careful consideration, we feel that it has merit but does not fully meet PLOS ONE’s publication criteria as it currently stands. Therefore, we invite you to submit a revised version of the manuscript that addresses the points raised during the review process.

You addressed statistical issues well after discussing your biostatistician. However, we are still worrying about overfitting issue from multivariable analysis with the small number of cohort. Our biostatistician recommends to build the model with n=30+30 as a discovery and then to validate with the smaller dataset. As for me, identification itself of circulating microRNA/isomiRs from enough number of samples can be acceptable without validation. Since you have performed pricey experiments with valuable human samples, more reasonable analysis would make your findings more convincing.

However, I know it would be very challenging to reanalyze all your data. Therefore, I recommend that the statistical issue should be described as the limitation of your study. Further, you would be better to provide a table to clarify the process of multivariable stepwise selection of your candidates.

We would appreciate receiving your revised manuscript by Apr 17 2020 11:59PM. To enhance the reproducibility of your results, we recommend that if applicable you deposit your laboratory protocols in protocols.io, where a protocol can be assigned its own identifier (DOI) such that it can be cited independently in the future. For instructions see: http://journals.plos.org/plosone/s/submission-guidelines#loc-laboratory-protocols

We look forward to receiving your revised manuscript.

Kind regards,

Hyun-Sung Lee, M.D., Ph.D.

Academic Editor

PLOS ONE

Reviewers' comments:

Reviewer's Responses to Questions

**Comments to the Author**

1. If the authors have adequately addressed your comments raised in a previous round of review and you feel that this manuscript is now acceptable for publication, you may indicate that here to bypass the “Comments to the Author” section, enter your conflict of interest statement in the “Confidential to Editor” section, and submit your "Accept" recommendation.

Reviewer #2: All comments have been addressed

2. Is the manuscript technically sound, and do the data support the conclusions?

Reviewer #2: Yes

3. Has the statistical analysis been performed appropriately and rigorously? 

Reviewer #2: Yes

4. Have the authors made all data underlying the findings in their manuscript fully available?

Reviewer #2: Yes

5. Is the manuscript presented in an intelligible fashion and written in standard English?

Reviewer #2: Yes

6. Review Comments to the Author

Reviewer #2: (No Response)

7. PLOS authors have the option to publish the peer review history of their article (what does this mean?). If published, this will include your full peer review and any attached files.

Reviewer #2: Yes: Tao-tao, Liu

---

## [Author Response · Author response to Decision Letter 1]

14 Mar 2020

Responses to the comments of Editor.

We gratefully appreciate your review our manuscript and the helpful suggestions. Our point-by-point responses to your comments and suggestions are listed below.

Comment 1

I recommend that the statistical issue should be described as the limitation of your study. 

As suggested by the editor, we also think that the method of statistics needs discussion. We appreciate you for accepting our method. Our laboratory is currently preparing for a prospective validation. We hope to report this result in the future. The following description has been added to the Discussion part. (‘‘Our results were also influenced by the number of samples assigned to each group and the method of statistical analysis.’’; line270-272 and ‘‘External validation is preferred to confirm accuracy of our results, but it is difficult because there is no public database containing information on circulating isomiR in ESCC patients. Therefore we tested the application for our diagnostic panel using another cohort, but it was using retrospective single institution samples after all. Prospective confirmation study is needed before clinical application.’’).

Comment 2

Further, you would be better to provide a table to clarify the process of multivariable stepwise selection of your candidates.

As suggested by the editor, we described selection method of variables for the Stepwise model in Material and methods part, and shown their results in Results part. These results are also shown in S1 Fig. and S3 File.

---

## [Editor Report · Decision Letter 2]

17 Mar 2020

Circulating microRNA/isomiRs as novel biomarkers of esophageal squamous cell carcinoma.

PONE-D-19-31398R2

Dear Dr. Tahara,

We are pleased to inform you that your manuscript has been judged scientifically suitable for publication and will be formally accepted for publication once it complies with all outstanding technical requirements.

With kind regards,

Hyun-Sung Lee, M.D., Ph.D.

Academic Editor

PLOS ONE
---

## [Editor Report · Acceptance letter]

23 Mar 2020

PONE-D-19-31398R2 

Circulating microRNA/isomiRs as novel biomarkers of esophageal squamous cell carcinoma. 

Dear Dr. Tahara:

I am pleased to inform you that your manuscript has been deemed suitable for publication in PLOS ONE. Congratulations! Your manuscript is now with our production department. 

With kind regards,

on behalf of

Dr. Hyun-Sung Lee 

Academic Editor

PLOS ONE